# Dual Effect of Chemo-PDT with Tumor Targeting Nanoparticles Containing iRGD Peptide

**DOI:** 10.3390/pharmaceutics15020614

**Published:** 2023-02-11

**Authors:** Gye Lim Kim, Byeongmin Park, Eun Hyang Jang, Jaeun Gu, Seo Ra Seo, Hyein Cheung, Hyo Jung Lee, Sangmin Lee, Jong-Ho Kim

**Affiliations:** 1College of Pharmacy and Bionanocomposite Research Center, Kyung Hee University, 26 Kyungheedae-ro, Dongdaemun-gu, Seoul 02447, Republic of Korea; 2Department of Regulatory Science, Graduated School, Kyung Hee University, 26 Kyungheedae-ro, Dongdaemun-gu, Seoul 02447, Republic of Korea

**Keywords:** photodynamic therapy, nanoparticle, tumor targeting peptide, iRGD, combination therapy

## Abstract

Nanotechnology, including self-aggregated nanoparticles, has shown high effectiveness in the treatment of solid tumors. To overcome the limitations of conventional cancer therapies and promote therapeutic efficacy, a combination of PDT and chemotherapy can be considered an effective strategy for cancer treatment. This study presents the development of tumor-targeting polysialic acid (PSA) nanoparticles for chemo-PDT to increase the cellular uptake and cytotoxic effect in cancer cells. Chlorin e6 (Ce6), a photosensitizer, and the iRGD peptide (sequence; cCRGDKGPDC) were conjugated to the amine of *N*-deacetylated PSA. They generate reactive oxygen species (ROS), especially singlet oxygen (^1^O_2_), and target integrin αvβ3 on the cancer cell surface. To offer a chemotherapeutic effect, doxorubicin (Dox) was assembled into the core of hydrophobically modified PSA by connecting it with Ce6; this was followed by its sustained release from the nanoparticles. These nanoparticles are able to generate ROS under 633 nm visible-light irradiation, resulting in the strong cytotoxicity of Dox with anticancer effects in HCT116 cells. PSA nanoparticles with the dual effect of chemo-PDT improve conventional PDT, which has a poor ability to deliver photosensitizers to cancer cells. Using their combination with Dox chemotherapy, rapid removal of cancer cells can be expected.

## 1. Introduction

Cancer has threatened the health of human beings for a long time, and the number of cancer patients increases every year. Cancer treatments include surgery, chemotherapy, and radiotherapy. Nevertheless, the therapies mentioned above have many limitations, such as invasiveness, low effectiveness, high toxicity for normal tissue and cells, and a short-half life. For a more effective treatment, photodynamic therapy (PDT) needs to be performed with other therapies [1,2]. PDT uses a photosensitizer, a photo-activated molecule, and molecular oxygen in the tissue to generate reactive oxygen species (ROS), especially singlet oxygen (^1^O_2_), under light irradiation [3,4,5]. ROS can be generated around the irradiated area to kill cells nearby the laser-affected part while avoiding the non-irradiated cells. Thus, the cytotoxic effect depends on the laser being on/off, and on its power. PDT has many advantages in terms of therapeutics and reduced side effects. However, the laser has trouble penetrating tissue and reaching the deeper region of the tumor [6]. Therefore, the dual strategy of carrying out both chemotherapy and PDT represents an improved therapy for cancer treatment [7,8].

Drug delivery systems using nanotechnology for cancer treatment have been studied and shown remarkable effectiveness over chemotherapeutic agents [9,10,11]. Biocompatible and biodegradable polymer nanoparticles can contain small-molecule drugs, and can conjugate with therapeutics or imaging agents [12]. The polymer structure protects loaded or conjugated drugs from blood clearance, and normal cells and tissues from the toxic agent. In addition, nanoparticles circulate in the bloodstream for a long time and accumulate in tumor tissue and cells through the enhanced permeability and retention (EPR) effect [13,14,15,16,17]. Self-aggregated nanoparticles of proper diameter can be adjusted to act as nanomedicines and drug-delivery carriers. Polysialic acid (PSA) can be applied as a biomaterial due to its good biocompatibility and biodegradability. The amine and carboxyl groups of PSA provide opportunities for chemical modification by linking with drugs, imaging agents, or peptides [12].

iRGD (sequence: cCRGDKGPDC) is a 9-amino acid peptide that tends to infiltrate into the tumor tissue [18]. The iRGD peptide homes in on and penetrates the tumor through integrin αvβ3 and neuropilin-1 (NRP-1) mechanisms. Integrin αvβ3 is a heterodimeric adhesion protein on the surface of cancer cells. After the RGD sequence motif of the peptide mediates binding to αvβ3, an intracellular protease is activated and cleaves the CendR (R/KXXR/K) of the peptide. Then, CendR can bind to NRP-1 and consequentially activate the endocytosis and transcytosis pathways. Additionally, the iRGD peptide can be spread far more extensively into extravascular tumor tissue than the RGD (arginylglycylaspartic acid) peptide [19,20,21,22,23]. In summary, conjugating the iRGD peptide could promote increased accumulation and penetration of anticancer agents into αvβ3-positive tumor tissue.

In this study, doxorubicin (Dox) was selected as a chemotherapeutic drug, and chlorin e6 (Ce6) was chosen as a second-generation clinically used photosensitizer [24,25]. Dox was loaded in hydrophobically modified PSA nanoparticles by conjugating Ce6 and the iRGD peptide with the amine residue of PSA (D@iNPs). The polymer structure consisted of the hydrophilic property of PSA and the hydrophobic property of Ce6, and this amphiphilic character led to the generation of self-aggregated and unified PSA nanoparticles. In αvβ3-positive HCT116 cells, iRGD/Ce6-conjugated PSA (iNPs) was taken up more than free Ce6 and Ce6-conjugated PSA (NPs). Under laser irradiation, Dox released from the D@iNPs and ROS generated from Ce6 amplified the cytotoxic effects on the tumor cells. After intravenous injection of D@iNPs into HCT116 tumor-bearing mice, the effect of the combination therapy was identified in these mice (Figure 1).

In summary, in this study, we offer three strong points of evidence for the use of PSA nanoparticles in cancer treatment. First, the iRGD peptide encourages nanoparticles to target αvβ3, as well as promoting cellular uptake in cancer cells. Second, ROS induced by visible-light irradiation can kill cancer cells through diverse mechanisms. Lastly, time-released Dox from the hydrophobic core of nanoparticles damages cancer cells. These overall approaches may potentially enhance cancer treatment.

## 2. Materials and Methods

### 2.1. Materials

Polysilaic acid (PSA, Colominic acid sodium salt, MW = approx. 30 kD), *N*-hydroxysuccinimide (NHS), 1-ethyl-3-(3-dimethylaminopropyl) carbodiimide (EDC), Triethylamine (TEA) and 3-(2-Pyridyldithio)propionic acid *N*-hydroxysuccinimide ester (SPDP) were purchased from Sigma-Aldrich (St. Louis, MO, USA). Chlorin e6 (Ce6) was purchased from Frontier Scientific Inc. (Logan, UT, USA). cCRGDKGPDC (iRGD) peptide was synthesized by Peptron (Daejeon, Republic of Korea). Doxorubicin hydrochloride (Dox) was obtained from Future Chem (Seoul, Republic of Korea). Dimethyl sulfoxide (DMSO) was obtained from Junsei Chemical (Tokyo, Japan). Cellulose membrane dialysis tubes (MWCO = 3.5 kD) and Cellulose membrane dialysis tubes (MWCO = 1 kD) were purchased from Spectrum Laboratories (Rancho Dominguez, MO, USA) and used during dialysis. Dulbecco’s modified eagle medium (DMEM) high glucose, Rosewell Park Memorial Institute (RPMI) 1640 medium and fetal bovine serum (FBS) were obtained from WelGENE Inc. (Daegu, Republic of Korea). Penicillin-streptomycin (PS) was purchased from Hyclone-GE Healthcare Bio-Sciences (Logan, UT, USA).

### 2.2. Preparation of Nanoparticles

*N*-deacetylation of PSA was conducted, as reported in previous study [12]. 10 M NaOH (2 mL), deionized water (DW) (6 mL), Thiophenol (200 μL), DMSO (30 mL) and PSA (100 mg) were mixed. The mixtures were incubated at 80 °C for 3 h and dialyzed against 0.01 M ammonium carbonate solution at 4 °C for 72 h. The dialyzed solution was rapidly frozen in liquid nitrogen and lyophilized.

Ce6-conjugated PSA (NP) was prepared by EDC-NHS reaction. All these reactions, including Ce6 as described below, were conducted in light-shaded conditions. Briefly, Ce6 (9.94 mg), EDC (4.79 mg, 1.5 eq) and NHS (2.88 mg, 1.5 eq) were added to DMSO (10 mL) and mixed for 3 h at room temperature to activate carboxyl group of Ce6. Then, *N*-deacetylated PSA (De-PSA, 100 mg) in DMSO (90 mL) was added in pre-activated Ce6 solution and mixed overnight at room temperature. The mixtures were put into a membrane dialysis tube (MWCO = 3.5 kDa) and dialyzed against MeOH:DW = 1:1 solution for 24 h, and DW for 48 h in order. The dialyzed solution was lyophilized, and green-colored NP was obtained [26]. The conjugation efficiency of Ce6 was quantified using a UV/vis spectrophotometer (UV-1650, Shimanzu). The Ce6 concentration was determined by measuring absorbance at 405 nm and referring to a standard curve of free Ce6 concentrations in the same 1% (*v*/*v*) DMSO/distilled water solution.

To prepare iNP, an SPDP linker was used to conjugate the *N*-terminal of iRGD peptide and amine in NP. SPDP (0.95 mg) and NP (20 mg) were dissolved in DMSO and mixed for 2 h with continuous shaking at room temperature. After reaction was finished, the mixture was dialyzed against DW for 48 h. The dialyzed solution was collected in a round bottom flask and 3% of 2-mercaptoethanol (Merck, Armstadt, Germany) was added and mixed for 30 min at room temperature for reduction of disulfide bond in pyridyldithil-activated NP. The mixture was dialyzed in DW and lyophilized, and then sulfhydryl-activated NP was obtained. iRGD (2.87 mg) was dissolved in PBS- ethylenediaminetetraacetic acid solution (EDTA, USB Co., Cleveland, OH, USA) (2 mL, 100 mM sodium phosphate, 150 mM NaCl, 1 mM EDTA, pH = 7.5) and mixed with SPDP (0.95 mg, dissolved in 200 uL of DMSO). After 2 h, the solution was dialyzed in the same way mentioned above (MWCO = 1 kD). The dialyzed solution was mixed with sulfhydryl-activated NP for 18 h and dialyzed in DW and lyophilized (iNP). The conjugation ratio of iRGD molecule was determined using UV-vis absorption at 280 nm. In brief, 1 mg of iNP was dissolved in 1 mL of distilled water/DMSO (1:1 *v*/*v*) cosolvent. The conjugation ratio of iRGD molecule was analyzed based on UV-vis absorption at 280 nm with standard curve of iRGD.

Dox∙HCl (1 mg), iNP (9 mg) and trimethylamine (Sigma-Aldrich, St. Louis, MO, USA) (1 μL) were mixed overnight in room temperature. The mixture was dialyzed in a membrane bag (MWCO = 3.5 kD) to load hydrophobic Dox into iNP (D@iNP). D@NP was prepared as same process. The loading contents of DOX were determined using a UV/vis spectrophotometer at 490 nm.

### 2.3. Characterization of Nanoparticles

Nanoparticles were dissolved in DW (0.5 mg/mL). Then, the mean diameter, polydispersity index (PDI) and zeta potential were characterized by dynamic light scattering (DLS) (90 Plus Particle Size Analyzer, Brookhaven Instruments Corporation, Holtsville, NY, USA) at 25 °C. The morphological shapes of the nanoparticles were directly determined using transmission electron microscopy (TEM) (CM-200, Philips, San Diego, CA, USA). Freshly prepared nanoparticles (1 mg/mL in DW) were dropped onto a 400-mesh copper grid and stained with 5% (*w*/*v*) uranyl acetate solution before imaging. UV-visible spectra of Ce6 and iNP (1.6 μg/mL of Ce6) dissolved in DMSO: DW = 1:1 solution were obtained by UV-1650PC UV-vis spectrophotometer (Shimadzu, Kyoto, Japan) to check the UV-vis peaks of Ce6 in iNP. To detect modified molecules, energy dispersive x-ray analysis (EDX) (Field Emission S-4700) (Hitachi, Tokyo, Japan) was performed on the functionalized nanoparticles. EDX spectra were measured at 20 hV accelerating voltage. To verify the serum stability of nanoparticles, iNP (0.5 mg/mL) dissolved in 0.2 μm syringe filtered FBS (10%) containing PBS was placed into a 50 mL amber tube and incubated in a shaking water bath at 37 °C, 150 rmp. The changes in mean diameter and PDI were measured for 96 h, every 24 h, using DLS.

The release of Dox from D@iNP and D@NP was conducted by dialysis. D@iNP and D@NP (2 mL, 1 mg/mL) were dissolved in PBS, placed into a dialysis tube (MWCO 1000) and tightly tied by PTFE tape. Then, the dialysis tube was submerged in 100 mL round bottom flask containing 18 mL of PBS (pH = 7.4). The flasks were shaken in a shaking water bath at 37 °C, 150 RPM. Some 100 uL of samples (*n* = 3) from each flask were measured at predesigned time intervals from the release medium and put back. The concentration of Dox was measured by fluorescence absorption (λEx max 470 nm, λEm max 585 nm). The experiment was repeated three times, using the same protocol mentioned above.

For evaluating quantitative ROS, a bleaching test using p-nitroso-N,N′dimethylaniline (RNO) (Sigma-Aldrich, St. Louis, MO, USA) was conducted to verify the amount of singlet oxygen generation between Ce6, NP and iNP in vitro. Ce6, NP and iNP were dissolved in PBS (pH = 9.0, 1% DMSO) containing 1.8 mM of L-Histidine (Sigma-Aldrich, St. Louis, MO, USA) and 18 uM of RNO. The samples were irradiated using a laser (671 nm, SDL-series, Shanghai Dream Laser Technology Co., Ltd., Shanghai, China) (50 mW/cm^−2^) in a dark room. The RNO absorbance was measured at 440 nm of each sample (*n* = 3) using a UV-vis spectrometer (Agilent 8453 UB-visible Spectroscopy System, Agilent Technology, Santa Clara, CA, USA).

### 2.4. In Vitro Cytotoxicity and Cellular Uptake

HCT116 (Human colorectal carcinoma) was purchased from American Type Culture Collection (Rockville, MD, USA), and HT29 (Human colorectal adenocarcinoma) was purchased from Korea Cell Line Bank (Seoul, Republic of Korea). HCT116 and HT29 cells were maintained in 10% FBS and 1% PS with added RPMI at 37 °C, under humidified 5% CO_2_.

The cytotoxicity of diverse nanoparticles in HCT116 cells was studied with an enhanced cell viability assay kit (EZ-Cytox; DoGenBio, Seoul, Republic of Korea). HCT116 cells (1.0 × 10^4^ cells/well) were seeded in 96-well plates and incubated overnight at 37 °C. After cell adhesion, the cells were treated with various concentrations of Ce6, Dox, NP, iNP, D@NP, and D@iNP dissolved in serum-free RPMI for 4 h. Then, the media were changed with serum-free RPMI. The plates with laser-on groups were irradiated by visible light (633 nm, 10 J/cm^2^), and those with laser-off groups were incubated in the dark. After laser treatment, all plates were placed in a cell incubator and stayed in the dark for 20 h. After removal of the medium from the wells, 100 μL of 10% EZ-Cytox solution containing serum-free RPMI was added to each well and incubated for 30 min at 37 °C. When the color of the well changed, the absorbance of the wells (*n* = 5) at 450 nm was measured by a microplate reader (Versa Max™; Molecular Devices Corp., San Jose, CA, USA). The O·D value of the formazan was calibrated by subtracting the O·D value of 100 μL of 10% PBS containing serum-free RPMI and treated in the wells of each group.

Flow cytometry was conducted to observe the cell uptake of iNP in HCT116 and HT29 cells. Both cells were seeded in 6-well plates and serum-starved overnight. iNP (5 μg/mL of Ce6) dissolved in serum-free RPMI was added, and cells were incubated for 4 h in the dark. After that, trypsin-EDTA (WelGENE Inc., Daegu, Republic of Korea) was added to harvest cells, and the collected cells were centrifuged at 1500 rpm for 5 min. The supernatant was gently removed, and the cell pellet was washed with PBS (pH = 7.4). Then, suspended cells were centrifuged at 1500 rpm for 5 min, and the washing process mentioned above was repeated three times. The cell solution was analyzed by fluorescence-activated cell sorting (FACS) cater-plus flow cytometry (Becton Dickinson Co, Heidelberg, Germany).

Fluorescence imaging was conducted to evaluate the difference in the cellular uptake of nanoparticles in HCT116 and HT29 in vitro. HCT116 (4.0 × 10^4^ cells/well) and HT29 (3.5 × 10^4^ cells/well) are seeded into 4-well chamber slides. After stabilizing, the cells were serum-starved for 4 h, and a serum-free medium containing Ce6, NP, and iNP (5 μg/mL of Ce6) was added into each well of the plate and incubated for 4 h in the dark. Then, the chamber was washed with PBS several times, and cells were fixed with 4% paraformaldehyde solution for 10 min and washed with PBS. After 4,6-diamidino-2-phenylindole (DAPI) (Invitrogen, Carlsbad, CA, USA) staining and washing the cells three times, HCT116 and HT29 cells were observed using 400× magnification on a Leica fluorescence microscope.

To observe the distribution and effect of various nanoparticles in the cells, 5 × 10^5^ HCT116 cells were seeded on 35 mm confocal dishes and incubated overnight. Then, Ce6, Dox, NP, iNP, D@NP, and D@iNP (3 μM of Ce6 and 1.8 μM of Dox) dissolved in serum-free RPMI were treated in HCT116, respectively and incubated for 3.5 h at 37 °C. After treatment, 30 μM of HDCF-DA was added to each dish, and those dishes were incubated for 0.5 h in the dark. After incubation, the medium was changed to fresh serum-free RPMI, and the plates were irradiated by visible light (633 nm, 10 mJ/cm^2^). Then, the medium was removed, and the dishes were washed with PBS several times, fixed with 2% paraformaldehyde solution, and washed with PBS. After the cells were fixed, samples were stained with DAPI for 10 min. The fluorescence images were obtained by a confocal laser microscope (Leica TCS SP8, Leica Microsystems GmbH, Wetzlar, Germany).

### 2.5. Western Blot Analysis

To analyze the amount of integrin alpha V beta 3 (αVβ3) in HCT116 and HT29, each cell was harvested from the plates by scraping the cells. Cells were pelleted using a centrifuge at 1500 rpm for 5 min at 4 °C. Then, pellets were resuspended in the lysis buffer (1% SDS, 100 mM Tris-HCl, pH 7.4) containing a 100× protease inhibitor cocktail (Complete, EDTA-free, Roche, Sydney, NSW, Australia). After 30 min incubation at 4 °C, the debris was removed by centrifugation for 10 min at 3000 rpm. Then, the concentrations of soluble proteins in the lysis buffer were performed by bicinchoninic acid (BCA) protein assay (Pierce, Rockford, IL, USA). The quantified samples were boiled for 5 min in the SDS-loading buffer. Then, 10 μg of proteins were separated by SDS-polyacrylamide gel (10%) and subsequently transferred onto PVDF membranes. The membrane was blocked with 5% bovine serum albumin (BSA) in TBST (50 mM Tris-HCl, 150 mM NaCl, 0.1% Tween 20, pH 7.4) for 1 h, and then the membrane was incubated with a blocking solution with rabbit anti-human integrin αVβ3 antibody for 12 h at 4 °C. Finally, the membrane was incubated with the anti-rabbit IgG-HRP antibody for 2 h, and αVβ3 bands were detected by an enhanced chemiluminescence (ECL) system. The band-signal intensity of integrin αVβ3 was quantified using ImageJ software.

### 2.6. In Vivo Antitumor Effect

All experiments with live animals were performed in compliance with the relevant laws and institutional guidelines of the Institutional Animal Care and Use Committee (IACUC) at the Korea Institute of Science and Technology (KIST), and institutional committees approved the experiments. HCT116 tumors (RGD-positive tumors) and HT29 tumors (RGD-negative tumors) were induced into 5-week-old male athymic nude mice on the left side of the flank by subcutaneous injection of 1.0 × 10^7^ cells. When the tumor size reached about 100 mm^2^, the D@iNP was administered into HCT116 and HT29 tumor bearing mice by intravenous injection (*n* = 3 per each experimental group). Their biodistribution and tumor accumulation was observed using the IVIS Lumina III in vivo imaging system.

### 2.7. Statistical Anlysis

The results were presented as mean SD or SEM, and statistical comparisons between groups were carried out using a one-way ANOVA followed by the Student’s *t*-test, using SigmaPlot version 10.0.

## 3. Results and Discussion

### 3.1. Characterization of Nanoparticles

The NPs were synthesized through the conjugation of hydrophobic Ce6, a photosensitizer, to the hydrophilic N-deacetylated sialic acid backbone. This amphiphilic conjugate self-assembled into spherical nanoparticles (NPs) with a diameter of about 220 nm (Figure 2A). For αvβ3-positive tumor targeting, iRGD was conjugated to the amine groups of the NPs via the SPDP linker. About 31 molecules of Ce6 and 27 molecules of iRGD were conjugated to 1 sialic acid polymer, based on UV–vis absorbance data after conjugation. The feed amount of Dox was 20 wt% of the NPs and iNPs, and the loading efficiency of Dox was 82.15% and 83.32% in the D@NPs and D@iNPs, respectively. The mean diameter of the iNPs was 219.8 ± 1.3 nm, which was not significantly different from the diameters of the NPs, D@NPs, and D@iNPs. The spherical morphologies of these particles were confirmed in the TEM images. The PDI measured by DLS for all nanoparticles indicated that the nanoparticles were uniform and well dispersed in PBS. Because of the negatively charged amino acids, such as aspartic acid, and reduced amine in the PSA structure, the zeta potential value of the iNPs (−32.15 ± 1.74 mV) was more negative than that of the NPs (−24.35 ± 0.63 mV). These results show that the iRGD peptide was successfully conjugated to the amine of PSA.

De-PSA presented as white, while the NPs and iNPs presented as green and dark green, respectively, due to the green color of Ce6. These color changes indicated that Ce6 was successfully conjugated with the PSA backbone in the NPs and iNPs. Upon measuring the UV–vis spectra, the absorption bands of Ce6 and the iNPs were confirmed. Free Ce6 exhibited a Soret band at 403 nm, a Q band at 661 nm, and another peak at 500 nm. At the same concentration of Ce6, the Soret band of the iNPs was at 403 nm as well. However, the Q band of the iNPs shifted to 671 nm, compared to that of Ce6 (661 nm). The peak of Ce6 at 500 nm was not identified in the iNP group (Figure 2B).

To evaluate the ROS generation capability of the NPs and iNPs, bleaching of RNO, a singlet oxygen sensor, was used [27]. Samples were mixed with RNO and L-histidine, and the depletion of absorbance at 440 nm was measured with a UV–vis spectrometer after laser irradiation. Ce6, NP, and iNP samples were irradiated at 3 min intervals of on/off repetition. After 15 min, the relative concentration of RNO significantly decreased (10.5%) for free Ce6. Due to the quenching effect of Ce6 in the nanoparticles, the bleaching effects of the NPs (39.4%) and iNPs (44.5%) were slightly weaker than those of Ce6. Ce6 molecules were placed into the hydrophobic core of the NPs and iNPs, close to one another. Thus, the quenching effect of Ce6 in the nanoparticles led to a decrease in the ROS generation of Ce6. Nevertheless, the NPs and iNPs had good singlet-oxygen-generating abilities (Figure 2C).

After the NPs and iNPs dissolved in PBS containing 10% FBS (0.5 mg/mL) were incubated at 37 °C, the size of the NPs and iNPs was observed via DLS measurement for 96 h, every 24 h. The size and PDI of the iNPs were not significantly changed, but the size of the NPs slightly increased depending on the time (Figure 2D and Appendix A). Despite these differences, both the iNPs and NPs are stable in serum in general.

To confirm the sustained release of Dox from the NPs and iNPs, the release profile of Dox from the NPs and iNPs was observed at 37 °C. Dox released from the D@NPs and D@iNPs achieved values of 67.4% and 65.3%, respectively. The analogous Dox release profiles of the two different nanoparticles demonstrate that iRGD contained in the PSA structure does not interrupt the release of Dox from the nanoparticles (Appendix A). These results revealed that the nanoparticles formed were of a uniform size in aqueous solution for a few days. In addition, both Ce6 and iRGD were conjugated well with PSA (iNPs), and the iNPs also showed characteristics of Ce6 and iRGD. These iNPs and NPs may be promising drug carriers as well.

### 3.2. In Vitro Cellular Uptake and Cytotoxicity

Before verifying the cellular uptake of Ce6, NPs, and iNPs, identification of the expression level of integrin αvβ3 in different cell lines was conducted using a Western blotting assay. This assay was carried out in the HCT116 and HT29 cell lines, which are human colon cancer cell lines. The Western blot bands showed that integrin αvβ3 was more expressed in HCT116 than in HT29 (Appendix A).

The cytotoxicity of various nanoparticles was estimated using a cell viability assay in the HCT116 cell line. Cell viability decreased according to the higher concentration of Ce6 under the laser. In particular, the cell viability of the D@iNPs was less than 10% at a 0.2 μg/mL concentration of Ce6 with a laser (Figure 3A). In contrast, there was no severe cytotoxicity in all groups without laser irradiation (Figure 3B). Dox-loaded nanoparticles killed more cancer cells than nanoparticles without Dox under visible light. The iNPs effectively damaged cancer cells compared to the NPs with and without Dox (Figure 3C). Therefore, cell death was induced through the phototoxic effect of Ce6 in the nanoparticles, and amplified by targeting the ability of iRGD and the release of Dox.

For a comparative study on the cellular uptake aspect of the iRGD-NPs according to the αvβ3 expression pattern, we selected an HCT116 cell line with high expression of the integrin αvβ3 in the two cell lines; the comparative study was conducted using a Western blotting assay. Before verifying the cellular uptake of Ce6, NPs, and iNPs, identification of the expression level of integrin αvβ3 was conducted by Western blotting assay. The Western blot bands showed that integrin αvβ3 was more expressed in HCT116 than in HT29 (Appendix A).

The cellular uptake test was conducted in HCT116 and HT29 cells via flow cytometry. The cells were incubated in serum-free media with the iNPs for 4 h, and Ce6 fluorescence was detected. The fluorescence intensity of Ce6 in the HCT116-treated iNPs was much higher than that of HT29. These results demonstrate that the nanoparticle-derived cellular uptake was rapid and lasted for a long time, and also that tumor-targeting peptides boosted the uptake of the nanoparticles into cancer cells. Further, the Ce6 fluorescence of the iNPs was detected more in HCT116 cells than in HT29 cells (Figure 3D).

Fluorescence imaging was also conducted to assess the cellular uptake of the nanoparticles. HT29 and HCT116 cells were both incubated in serum-free media with Ce6, NPs, and iNPs for 4 h. As expected, the fluorescence intensity of the iNP group in HCT116 was higher than that of the NP and free Ce6 groups. Compared to the cell lines treated with the iNPs, the fluorescence in the HCT116 group was brighter than that in the HT29 group. These images support the findings that more integrin αvβ3 was expressed in HCT116 cells, and that iRGD peptides contributed to the uptake of nanoparticles into the cells and provided cancer-targeting ability to the nanoparticles (Figure 3E).

The in vitro cellular uptake, Dox release, and intracellular ROS generation in HCT116 cells were observed following laser irradiation via confocal imaging. All samples apart from the untreated group and the Dox group were irradiated by 633 nm visible light (10 J/cm^2^) after 2.5 h of nanoparticle or drug treatment and 0.5 h of HDCF-DA treatment. HDCF-DA was treated with serum-free RPMI, owing to the cellular esterase activity. Under the light, Ce6 activated to induce ROS generation, and the Dox released from the nanoparticles moved to the nucleus and damaged the cells. Thus, Ce6 and Dox converted HDCF-DA to DCF, producing green fluorescence. In comparison with the nanoparticles without iRGD (NPs and D@NPs), the cells treated with the iRGD-conjugated nanoparticles (iNPs and D@iNPs) exhibited enhanced cellular uptake in HCT116 cells, as shown by the Ce6 fluorescence (red). The damaged nuclei of cells were observed through the DAPI-stained nuclei (blue) in the cells treated with the Dox-containing nanoparticles such as the D@NPs, and especially the D@iNPs, due to the dual cytotoxic effect of doxorubicin and ROS generated by Ce6. These results suggest that treatment with PDT using Ce6 and chemotherapy using Dox can kill cancer cells effectively (Figure 4).

### 3.3. In Vivo Antitumor Effect

For a precise analysis of active tumor-targeted delivery of the D@iNPs, we compared the biodistribution of the D@iNPs in HCT116 and HT29 tumor-bearing mice. After intravenous injection of the D@iNPs, the time-dependent fluorescence changes at the tumor site were significantly different between HCT116 and HT29 tumors (Appendix A). As shown in the in vivo analysis, the accumulation of D@iNPs increased by about two-fold in the case of HCT116 tumors with a high expression of the integrin αvβ3 group. This result signifies that binding between iRGD and integrin αvβ3 was responsible for the increased accumulation of D@iNPs in the HCT116 tumor tissue of the mice. To verify the therapeutic efficacy of the D@iNPs, HCT116 tumor-bearing mice were injected with nanoparticles (NPs, iNPs, D@NPs, D@iNPs). The other groups of mice were treated with Ce6 and Dox to compare with the nanoparticles. After monitoring the tumor volumes for 15 days, the change in tumor sizes indicated that the D@iNPs inhibited the increasing tumor size the most effectively. After comparing the excised tumors, the size of tumors treated with the D@iNPs was the smallest (Figure 5A). Upon observing tumors on the right flank of the mice on day 15, there was a significant increase in the Ce6 and Dox groups in the tumors, similar to the control group (Figure 5B). The body weights of all groups remained constant, showing that the nanoparticles do not have serious toxicity (Figure 5C).

In addition, evaluation of the synergy of the combination therapy was performed with the Bliss independence model, which is one of the most popular models to calculate the combination index (CI, CI = (E_A_ + E_B_ − E_A_E_B_)/E_AB_, where E_A_ and E_B_ represent the observed effects of therapies A and B, respectively, and E_AB_ is the effect of A combined with B) [27]. The CI was 1.07 in the D@NPs and 0.86 in the D@iNPs, respectively, meaning that the D@iNPs demonstrated synergy; a combination can be considered to have synergy when the CI is below 1 [28]. Therefore, the D@iNPs had a greater antitumor effect than the other nanoparticles because they had a tumor-targeting ability and the dual effect of PDT and chemotherapy.

The antitumor effect was assessed histologically by Ki67 staining. The Ki67 was significantly negatively expressed in the tumor section of the D@iNP group, which implied that D@iNP most effectively decreased the proliferation rate (Figure 5D). Hematoxylin and eosin (H&E) staining was also performed to evaluate tumor tissue damage. The tumor of the D@iNP group had the most severe damage among the other groups (Figure 5E). These results suggested that D@iNP had a synergistic antitumor effect. 

## 4. Conclusions

In this study, we synthesized PSA nanoparticles (D@iNPs) conjugated with a tumor-targeting peptide (iRGD) and a photosensitizer (Ce6) and loaded a chemotherapy drug (Dox). The nanoparticles were well dispersed in the aqueous solution, generating the EPR effect. The ability to generate ROS and the sustained release of Dox were satisfactory. In vitro, these nanoparticles were taken up more in the αvβ3-expressed cancer cells. The D@iNPs induced high ROS generation upon visible-light irradiation and killed the cancer cells by inducing the dual effect of PDT and chemotherapy. In vivo, the D@iNPs inhibited the growth of the HCT116 tumors effectively through the synergistic effects of specific tumor-targeting ability and chemo-PDT, without toxicity to the mice. In conclusion, D@iNPs could potentially be an antitumor therapy.

## Figures and Tables

**Figure 1 pharmaceutics-15-00614-f001:**
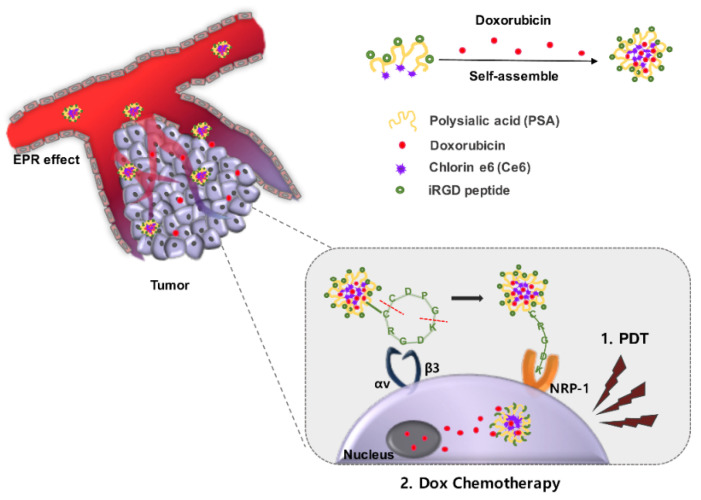
Schematic illustration of chemo-PDT nanoparticles. The mechanism of the dual effect of chemo-PDT with tumor-targeting nanoparticles containing the iRGD peptide. Formation of self-aggregated nanoparticles and Dox loading into the hydrophobic core in water-based solutions. After the iNPs accumulate in the tumor through the EPR effect, CendR binds to αvβ3 and is cleaved by an intracellular protease. According to the cut-off recognized by NRP-1, endocytosis and transcytosis are stimulated. Dox released from nanoparticles affects the nucleus, and Ce6 generates ROS through the laser, eventually killing the cells.

**Figure 2 pharmaceutics-15-00614-f002:**
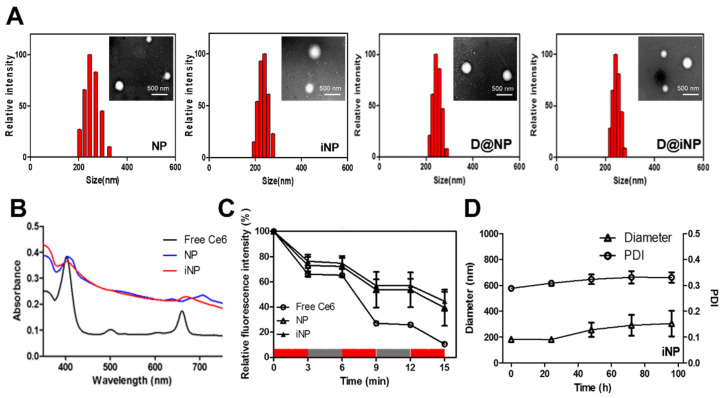
Characterization of nanoparticles. (**A**) Size distribution of NPs, iNPs, D@NPs, and D@iNPs measured by dynamic light scattering (DLS). The insets show TEM images. (**B**) UV–vis spectra of free Ce6 (black line), NPs (blue line), and iNPs (red line) at 1.6 μg/mL of Ce6. (**C**) ROS generation of free Ce6, NPs, and iNPs depending on the laser on (red)/off (grey) (50 mW/cm^2^). (**D**) Serum stability of iNPs for 96 h.

**Figure 3 pharmaceutics-15-00614-f003:**
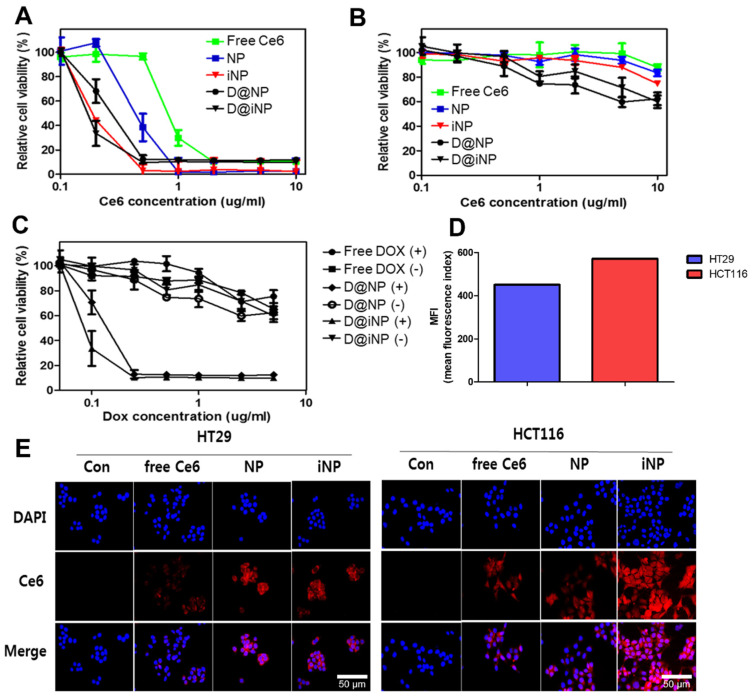
In vitro cytotoxicity and cellular uptake of nanoparticles. CCK assay of Ce6, Dox, NPs, iNPs, D@NPs, and D@iNPs with light and without light (*n* = 5) (10 J/cm^2^). (**B**) Ce6 concentration depended on cell viability with (**A**) and without light (**B**). (**C**) Dox concentration depended on cell viability with and without a laser. (**D**) Flow cytometry analysis of the cellular uptake 24 h after iNP treatment in HT29 and HCT116 cells. (**E**) Fluorescence images of the cellular uptake of Ce6, NPs, and iNPs after 4 h treatment in HT29 and HCT116 cells. Scale bar = 50 μm.

**Figure 4 pharmaceutics-15-00614-f004:**
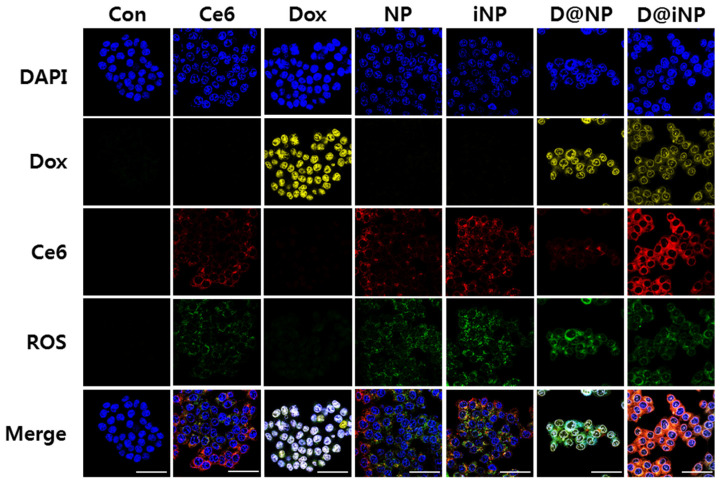
In vitro cellular uptake, Dox release, and ROS generation of the nanoparticles in HCT116 cells; DAPI (blue), Dox (yellow), Ce6 (red), and ROS (green). Cells were treated for 3.5 h with drugs or nanoparticles, and for 0.5 h with HDCF-DA. Scale bar = 50 μm.

**Figure 5 pharmaceutics-15-00614-f005:**
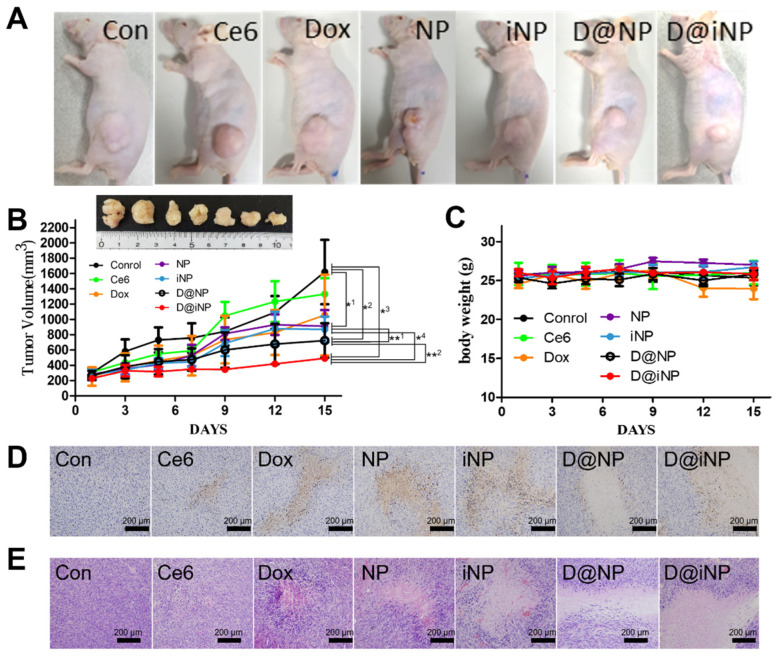
In vivo photodynamic therapeutic efficacy in HCT116-bearing mice. (**A**) Tumor growth curves of mice for 15 days after treatment with saline, Ce6, Dox, NPs, iNPs, D@NPs, and D@iNPs. (**B**) Body weights of mice for 15 days after treatments. ** and * indicate differences at the *p* < 0.01 and <0.05 significance levels, respectively, analyzed using one-way ANOVA (*^1^: control vs. NPs or iNPs; *^2^: control vs. D@NPs; *^3^: control vs. D@iNPs; *^4^: NPs or iNPs vs. D@iNPs; **^1^: NPs or iNPs vs. D@NPs; **^2^: D@NPs vs. D@iNPs). (**C**) Images of tumor-bearing mice on day 15 after treatments. (**D**) Ki67 staining of tumors after treatments. (**E**) H&E staining of tumors after treatments. Scale bar = 200 μm.

## Data Availability

Not applicable.

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
