# Peer review of "Dual Effect of Chemo-PDT with Tumor Targeting Nanoparticles Containing iRGD Peptide"

_pharmaceutics, 2023, doi:10.3390/pharmaceutics15020614_

Round 1

Reviewer 1 Report

The paper reportes the chemo-PDT dual effect of nanoparticles based on iRGD peptides and some common drugs, such as Ce6 and Dox. The preparation, characterization, in vitro and in vivo activities are presented.

In my opinion, the research is not very innovative and offers few new or interesting ideas for such fields. The introduction to the paper is vague and superficial. Lots of papers have reported the combination of Ce6 and Dox. What are the innovations of the present study compared to other studies involving Ce6 and Dox?

The construction of nanocomposites is not innovative and the inherent connection among the usual drugs is not persuasive.

The characterization of the nanoparticles is not complete and it is necessary to present the morphology of NPs by using TEM or SEM.

The western blot results should be presented in context.

The picture 3 is less clear.  

    Also, the writing is hard to follow.

For example, the sentence “This study suggests a development of tumor targeting polysilalic acid” in Abstract (line 14) is strange.

“Measuring the UV-vis spectra, the absorption bands of Ce6 and iNP were confirmed.”

Therefore, I do not recommend the publication of this paper in the journal Pharmaceutics. The paper requires extensive revision before publication.  

Author Response

Please see the attachment including author's reply and revised manuscript.

The reviewers’ comments were carefully studied and reflected in the revised manuscript. The authors appreciated valuable comments. Also, we added the commented data in revised manuscript. We sincerely hope that these corrections can answer the reviewers concerns and meet the standard and the guideline of Pharmaceutics.

Reviewer 2 Report

The manuscript entitled “Dual effect of chemo-PDT with tumor targeting nanoparticles containing iRGD peptide” reported an iRGD peptide modified nanoparticle for tumor targeted combination therapy. Overall, such a nanomedicine exhibited no obvious advantageous compared with traditional drug delivery systems. Some specific comments were listed as below:

1. The expression of integrin in HCT116 and HT29 cells should be evaluated to confirm the targeting mechanism.

2. Serum-free medium was employed to investigate the cellular uptake behaviors. A relative description should be added?

3. The morphology of D@iNP should be measured by TEM.

4. The loading contents of DOX and Ce6 should be measured.

5. In Figure S1B and Figure 4, DOX could be released in vitro and in living cells. What`s the drug release mechanism?

6. The synergistic mechanism between DOX and Ce6 should be discussed to highlight the importance of this work.

7. The tumor targeting behaviors should be provided to confirm the active targeting abilities of D@iNP.

Author Response

(The authors gave the same response as above.)

Reviewer 3 Report

In this manuscript,

 Gye Lim Kim and colleagues, researched the title “Dual effect of chemo-PDT with tumor targeting nanoparticles 2 containing iRGD peptide” I would like to recommend it to be published in the Journal of Pharmaceutics if the below referenced concerns could be addressed.

(1)  How do confirm the loading capacity of hydrophobic Dox into iNP (D@iNP)?

(2)  In vitro cytotoxicity, the authors used two cell lines, but the results of this In vitro cytotoxicity did not mention the cytotoxicity of each cell line.

(3)  In drug released study, the release rate of drug from both nanoparticles were not significantly different. Please explain the effect of drug carriers on release of drug.

(4)  The figures of fluorescence images of the cellular uptake and in vitro cellular uptake were not clear and scale bars were not included in the figure.

(5)  For Serum stability of iNP, did you observe only your final formulation? I think comparison study should also performed for confirmation.

Author Response

(The authors gave the same response as above.)

Reviewer 4 Report

This manuscript describes nanoparticles based on iRGD and Ce6 PSA conjugates with excellent tumor growth inhibition. However, despite the excellent tumor growth inhibitory effect, supplementary results are required to support them. The manuscript will be available for publication after the following supplementary procedures have been completed.

Author Response

(The authors gave the same response as above.)

Round 2

Reviewer 1 Report

The authors have made important revisions. The paper can be accepted in the present form. 

Reviewer 2 Report

The author have addressed most of my concerns, and I have no more questions.

Reviewer 4 Report

This manuscript has been appropriately edited to reflect previous comments.